# Impact of a Congested Match Schedule on Internal Load, Recovery, Well-Being, and Enjoyment in U16 Youth Water Polo Players

**DOI:** 10.3390/sports13090286

**Published:** 2025-08-25

**Authors:** Andrea Perazzetti, Arben Kaçurri, Masar Gjaka, Marco Pernigoni, Corrado Lupo, Antonio Tessitore

**Affiliations:** 1Department of Movement, Human and Health Sciences, University of Rome ‘Foro Italico’, 00135 Rome, Italy; 2Sports Research Institute, Sports University of Tirana, 1001 Tirana, Albania; akacurri@ust.edu.al; 3Department of Sport and Movement Science, University for Business and Technology, 10000 Pristina, Kosovo; masar.gjaka@ubt-uni.net; 4Department of Coaching Science, Lithuanian Sports University, 44221 Kaunas, Lithuania; marco.pernigoni@lsu.lt; 5Neuro Muscular Function Research Group, School of Exercise & Sport Sciences, Department of Medical Sciences, University of Turin, 10124 Turin, Italy; corrado.lupo@unito.it

**Keywords:** recovery status, wellbeing, youth water polo, monitoring matches

## Abstract

This study aimed to monitor internal load, well-being, and recovery status in U16 male water polo players during a congested competitive period. Fourteen athletes from an Italian club played 12 matches (seven friendly, five official) over 7 days. The internal match load was measured using the session-RPE method (s-RPE). Perceived enjoyment was measured with the Enjoyment Likert Scale (ENJ), while recovery and well-being were evaluated using the Perceived Recovery Scale (PRS) and the Hooper Index (HI), respectively. No significant main effects were found on s-RPE, PRS, and HI considering friendly and official matches. However, ENJ was significantly higher during official matches (*p* < 0.005). PRS values were significantly affected by daytime matches (*p* < 0.005), with better perceived recovery reported for morning matches. Linear mixed model analysis revealed significant associations between s-RPE and HI (*p* = 0.001), the fatigue item (*p* = 0.001), and the PRS (*p* = 0.004). These results suggest that as internal load increases, players experience higher fatigue and report lower recovery and well-being scores. Employing simple, non-invasive tools like the RPE, PRS, and HI can help coaches and support staff to identify early signs of overtraining or insufficient recovery, allowing for more individualized load management and injury prevention in youth water polo athletes.

## 1. Introduction

In team sports, a congested match calendar has become a significant concern for practitioners, players, and sport organizations [1]. Nowadays, elite team sports players usually encounter congested schedules characterized by numerous matches within very short timeframes, both during the preseason and the competitive season. During this period, teams may also participate in multiple competitions concurrently, such as national and continental events. Research suggests that congested fixtures can impact on acute performance and well-being, limiting players’ availability in various team sports [2]. The increasing frequency of fixtures has intensified discussions around effective player load management, recovery capacity, and injury risk [3]. This issue highlights the challenge posed by extended competitive schedules, which often exceed traditional periodization strategies aimed at optimizing performance and preventing overtraining [4]. To address these issues, sports teams are increasingly utilizing innovative strategies like real-time load monitoring through GPS and wearable technology, along with machine learning and artificial intelligence models [5].

In team sports, several studies have observed changes in players’ training responses, through their perceived load and well-being status during regular and congested weeks. Clemente et al. [6], studying professional futsal players, observed greater session-RPE scores on matchday-3 (i.e., 3 days before a match, MD-3) in regular weeks compared to the same day in congested weeks, during which MD-3 is the day immediately after a match. Similar trends were also reported by the same author for professional handball [7], with greater scores in training loads and reduced scores in well-being during match MD-2 and MD-3 in normal weeks. Furthermore, a study on elite volleyball players [8] found greater session-RPE scores in MD-1 (i.e., 1 day before a match) in regular weeks compared to congested ones. Overall, these findings suggest that during congested weeks, coaches tend to prioritize recovery and match preparation over high training loads. Conversely, during non-congested periods, coaches are freer to adopt a tactical periodization approach to training.

The literature on team sports has highlighted the impact of congested matches on players, who often participate in over 60 official matches annually. These players often experience residual fatigue and have greater injury risks when resting for less than 3 or 4 days between matches compared to more than 6 days of recovery [9]. A longitudinal study by Chmura et al. regarding soccer found that players tended to accumulate greater fatigue in the later stages of the season, particularly when participating in additional tournaments like the national cup, the Champions League, or the Europa League [10]. In basketball, Conte et al. [11] found that players reported higher perceived fatigue and lower well-being on the second game day when playing two games per week on Friday and Saturday. In soccer, a prospective six-season study by Carling et al. [12] found a significantly greater risk of sustaining an injury in the final 15 min of play in the second match and in the first half of play in the third match during three consecutive matches played within a ≤4-day period compared to matches played outside the congested cycle.

Although a considerable amount of research has explored the effects of congested training periods in team sports, water polo has received limited attention regarding how this issue influences coaches’ behavior and players’ performance, recovery, and injury risk [13]. In major international water polo competitions, fixtures are typically scheduled within a few days, for example, eight matches over 15 days during the Olympic Games or national team tournaments, or three matches over four days in club competitions such as the Final Four of the Champions League and the Euro Cup. A study by Botonis et al. [14] investigated the wellness and perceived internal training load of elite Greek water polo players over four weeks leading up to the playoff phase. The period included an overloaded training phase (weeks 1 and 2), followed by a reduced training load phase (weeks 3 and 4). The results showed that the increased training load during the overloaded phase negatively affected wellness scores but did not significantly compromise sport-specific performance. Conversely, the subsequent reduction in training volume and intensity improved both wellness scores and sport-specific performance metrics by the end of the intervention, compared to the baseline and the end of the overload period. Brisola et al. [15] conducted a study on female water polo players, comparing a two-week training load intensification strategy between groups of functional and non-functional overreaching players. The findings indicate that training intensification followed by tapering can improve performance more effectively than standard training methods. Nonetheless, it may also result in functional impairment and diminished performance. Consequently, constant monitoring of performance, particularly during intensified periods, is essential to prevent negative effects. In this sense, the increased competitive commitments require frequent travel across different time zones, leading to fatigue from jet lag and reducing the time available for recovery [16]. A survey by Barrenetxea-Garcia [17] highlighted the need for tailored recovery interventions that consider the specific needs of water polo players and coaching staff. It revealed a gap between the perceived importance of recovery strategies and their actual use. Coaches play a pivotal role, but their limited understanding may lead to ineffective practices, increasing the risk of overtraining and injuries while passing on incorrect information to athletes.

In water polo, considering that preparation for significant international tournaments involves a high-volume, intensified training phase followed by a series of competitive unofficial matches designed to improve players’ competitive readiness, the increased workload may impact their recovery. A study of Botonis et al. [18] examined the impact of international tournament participation on heart rate variability and perceived recovery in elite water polo athletes. Heart rate variability was assessed over five consecutive days during an intensified pre-tournament training phase and for three subsequent days while the team participated in three international matches. The results indicated that, under the accumulated fatigue and residual stress effects, perceived recovery and internal workloads were diminished on tournament days compared to the pre-tournament period (intensified training phase), suggesting that additional factors may influence the overall perception of recovery among subjects. In this context, ref. [19] examined the impact of intensified training on sleep activity, hormonal responses, and overall wellness in elite water polo players preparing for the Tokyo Olympics. Their findings revealed that substantial increases in training load during camp led to significant sleep disturbances and elevated salivary cortisol levels, both of which normalized after the camp. These results suggest that an intensified workload, coupled with insufficient recovery, may disrupt sleep patterns and potentially elevate the risk of infection. Barrenetxea-Garcia et al. [20] addressed the importance of recovery in a global survey on sleep quality, revealing that both senior and youth international water polo players reported poor sleep in terms of both quantity and quality. They frequently failed to meet recommended sleep durations and showed significant variability in sleep efficiency, especially during training camps and competitions. However, the limited research on congested fixtures in water polo primarily focuses on adult players. Except for one study examining training load distribution in a young female team, specifically its effects on upper respiratory tract infection symptoms and muscle damage markers [21], there is a lack of research on the impact of congested training periods in youth water polo. For this reason, the aim of the present study was to investigate the players’ perceptions of effort, enjoyment, recovery, and well-being during a congested schedule that included both official and training matches in male youth water polo players. Based on this, we hypothesized that players’ internal load (s-RPE), well-being (HI), perceived recovery (PRS), and enjoyment (ENJ) would be influenced by contextual match factors such as the time of day (morning vs. afternoon matches), the match type (friendly vs. official), and the match day sequence within the congested schedule, and that higher s-RPE would be associated with reduced PRS and HI values in the following days.

## 2. Materials and Methods

### 2.1. Subjects

Fourteen Italian U16 male water polo players (mean ± SD: age 15.4 ± 0.7 years; body mass: 67.8 ± 7.8 kg; body height: 175.6 ± 4.9 cm; body mass index: 22.0 ± 2.7; playing experience: 6.1 ± 1.2 years), all from the same club, were recruited for this study.

Informed consent was obtained from players’ parents (guardians) after they were informed of the study’s main objectives. This study was conducted in accordance with the ethical principles of the 1964 Declaration of Helsinki and was approved by the local ethics committee of the University of Rome ‘Foro Italico’ (CAR 99/2021).

Throughout the season, each player participated in regular training, matches, and tournaments, and the players belonged to the following playing positions [22]: perimetral players (*PP*, n = 8); center defenders (CD, n = 3); center forwards (CF, n = 3); and goalkeepers (GK, n = 2). Playing positions were not used for statistical analyses and are listed only to provide more detailed descriptive information regarding participant players. Furthermore, for each player, the following inclusion criteria were used for the final analysis: (i) participating in all matches during the study period for at least the mean total duration of a quarter (15 min per match) [23]; (ii) not presenting any injury or illness that could impair matches’ performance; and iii) exclusion of goalkeepers from the analytical sample, as their physical demands differ significantly from those of field players.

### 2.2. Experimental Design

This study monitored and assessed the internal load, perceived recovery, well-being, and enjoyment of youth male water polo players during a congested match period consisting of 12 matches over 7 days. The matches were played between 18 and 29 July 2023, comprising 7 friendly matches over 4 days, followed by 1 day of rest, and then 5 official matches over the next 3 days. After each match, individual RPE scores were collected by asking players, ‘How intense was your session?’ using the Italian translation of the Borg category-ratio 10 scale (CR-10), as modified by Foster et al. [24]. The match internal load was then calculated using the session-RPE (s-RPE) method [25,26]. The use of the s-RPE method in youth athletes has already been validated in youth water polo players [27]. Simultaneously, players’ perceived enjoyment was assessed using the Italian translation and adaptation of the Exercise Enjoyment Scale (ENJ) [23,28,29]. Every morning, the well-being status was monitored using the Hooper Index [30], while recovery was assessed using the 10-point Perceived Recovery Scale (PRS) [31] before each match. To minimize variability in individual recovery and well-being, players refrained from using any recovery interventions and were instructed to maintain consistent lifestyles and nutritional habits aligned with those of their teammates throughout the entire study period. Players’ internal match load, enjoyment, perceived recovery, and well-being were analyzed in relation to (i) the playing time, defined as the total minutes played by each athlete in all matches; (ii) the match time of day, distinguished by matches played in the morning vs. matches played in the afternoon in the same day; (iii) the match type, distinguished by friendly matches (FMs) and official tournament matches (TMs); and (iv) the match day, representing the chronological days of matches played.

### 2.3. Procedures

Players became thoroughly familiar with the RPE, ENJ, PRS, and HI scales during the three weeks leading up to the start of this study. Throughout this period, they received both printed and digital copies (individually via WhatsApp) of the questionnaires each day, with instructions to read and interpret the scales before and after every training session, friendly match, and official match.

During the data collection period (starting 3 days before the first match and ending 3 days after the last match), players’ RPE and ENJ responses were recorded within 30 min after the end of the match, before players left the swimming facility. To minimize potential post-match interference and standardize the timing of responses, players were trained to use their smartphones for online data collection, which has been shown to be a reliable tool among athletes [32]. Players completed the RPE and ENJ scales using a customized Google Forms questionnaire, which mirrored the content of the printed and digital scales provided during the familiarization period. The questionnaire was sent to their personal smartphones via WhatsApp. Using the session-RPE (s-RPE) method [26], the internal match load was calculated by multiplying each player’s RPE score by the number of minutes played, with the result expressed in arbitrary units (a.u.). For the ENJ scale, players rated their match enjoyment on a 7-point scale (1 = not at all, 7 = extremely enjoyable).

Perceived recovery was assessed using the 10-point Perceived Recovery Scale [31], ranging from 0 (‘very poorly recovered/extremely tired’) to 10 (‘very well recovered/highly energetic’). This scale has previously been used in other team sports [33,34] as well as in water polo [35]. Specifically, the PRS was administered 36 h post-match (PRS_36h_, when there was a rest day after the last match, in the morning between 8:00 and 8:30), 12 h post-match (PRS_12h_, in the morning between 8:00 and 8:30), and 6 h post-match (PRS_6h_, in the afternoon between 17:30 and 18:00). Players’ well-being was monitored using the Hooper Index [30], a self-assessment questionnaire based on four subjective measures, each rated on a 7-point Likert scale. These measures were administered daily at the same time as the morning PRS (between 8:00 and 8:30) and assessed the athletes’ perceived levels of (i) stress, (ii) fatigue, (iii) muscle soreness (DOMS), and (iv) sleep quality. Scores for stress, fatigue, and DOMS ranged from 1 (‘very, very low’) to 7 (‘very, very high’), while sleep was rated from 1 (‘very, very bad’) to 7 (‘very, very good’). The overall Hooper Index score was obtained by summing up the four individual ratings.

Table 1 shows the scheduling of all matches played and the administration of the monitoring questionnaires.

Anthropometric measurements included body mass and height. Body mass was assessed using a Tanita SECA 762 scale (measuring range: 0–150 kg; SECA, Hamburg, Germany), while height was measured with a SECA 213 stadiometer (measuring range: 20–205 cm; SECA, Hamburg, Germany).

### 2.4. Statistical Analysis

Descriptive statistics for all variables are presented as the mean ± standard deviation (SD). To assess the effects of contextual factors on players’ internal load, perceived recovery, well-being, and enjoyment, separate linear mixed-effects models were used. Linear mixed models were used to assess how various contextual factors affected each outcome: s-RPE, ENJ, PRS_12h_ and PRS_6h_, HI, and its individual components (fatigue, sleep, stress, and DOMS). The fixed effects included in the models were as follows: (i) the match type (friendly match (FM) vs. tournament match (TM)); (ii) the match time of day (morning vs. afternoon, for comparison between PRS_6h_ and PRS_12h_); and (iii) the match day (chronological progression of the congested match schedule (day 1 to day 7)). Players were included as a random factor to account for repeated measures. For each significant main effect, the effect size was calculated using Cohen’s d, with the following interpretation: 0.2, trivial; 0.2–0.6, small; 0.6–1.2, moderate; 1.2–2.0, large; and 2.0, very large [36].

In addition, separate linear mixed models between continuous variables were used to assess the associations between s-RPE and PRS_12h_ and well-being indicators (HI, fatigue, sleep, stress, and DOMS). In this analysis, the intraclass correlation coefficient (ICC) was calculated to assess the consistency of the measurements.

All results of the main effect analyses are reported as F and *p* values. Furthermore, 95% confidence intervals (CI) were computed and reported for estimated marginal means to enhance interpretability and indicate the precision of the effects.

All statistical analyses were performed using the Jamovi software (version 2.6.44, x64; Jamovi, Sydney, Australia), and the statistical significance was set at *p* < 0.05.

## 3. Results

Table 2 presents the descriptive statistics (mean ± SD) for perceived exertion (RPE), enjoyment (ENJ), session-RPE (s-RPE), and players’ playing time during friendly (FM) and tournament (TM) matches, respectively.

The average players’ PRS value was 4.5 ± 2 and 3.1 ± 2 for PRS_12h_ and PRS_6h_, respectively, while it was 4.0 ± 2.1 for PRS_12h_ after friendly matches and 3.8 ± 2.1 for PRS_12h_ after tournament matches (Figure 1). Regarding the match time of day as a contextual factor, the linear mixed model revealed a significant main effect only on the PRS (F = 34.7; *p* < 0.001; estimate = −1.432; mean difference ± standard error [PRS_12h_ − PRS_6h_] = 1.42 ± 0.32; 95% CI = 2.88, 4.48; ES = 0.703, moderate), while no main effects were observed for the s-RPE (F = 0.02; *p* = 0.893) and the ENJ (F = 2.54; *p* = 0.113).

This study found no main effects on the s-RPE, PRS_12h_, or HI when considering friendly and tournament matches. However, the ENJ was significantly influenced by the type of match (F = 9.2; *p* = 0.003, estimate = −0.742; mean difference ± standard error [ENJ_FM − EN_TM] = 0.68 ± 0.26; 95% CI = −1.22, −0.26; ES = 0.396, small) (Figure 2).

The analysis of the linear mixed model also revealed that the match-day factor had significant main effects on the HI (F = 4; *p* = 0.002; estimate = 13.89; 95% CI = 11.48, 14.79), sleep (F = 2.89; *p* = 0.0014; estimate = 3.47; 95% CI = 2.86, 3.65), PRS12h (F = 2.7; *p* = 0.02; estimate = 4.76; 95% CI = 4.30, 6.40), and the daily s-RPE (F = 16.6; *p* < 0.001; estimate = 311.1; 95% CI = 666; 865.5). In this case, given the large number of pairwise comparisons, effect sizes were not reported.

Based on the pooled match data, the linear mixed model analysis between continuous variables revealed significant relationships between s-RPE and the Hooper Index (HI) (F = 11; *p* = 0.001; estimate = 0.001; 95% CI = 11.86, 15,92; ICC = 0.798), the fatigue item (F = 11; *p* = 0.001; estimate = 0.002; 95% CI = 3.29, 4.47; ICC = 0.729), and the PRS_12h_ score (F = 9.01; *p* = 0.004; estimate = −0.003; 95% CI = 3.93, 5.59; ICC = 0.562).

Figure 3 illustrates the s-RPE and HI trends over the data collection period, starting three days before the congested match schedule and continuing until three days after its end.

## 4. Discussion

The main purpose of this study was to examine male youth water polo players’ perceptions of effort, enjoyment, recovery, and well-being factors during a congested match schedule that included both official and training matches. To the best of our knowledge, this is the first water polo study that specifically investigates matches played twice in the same day (one match in the morning and one in the afternoon) and repeatedly over consecutive days (12 matches in 7 days). This kind of scheduling provides a valuable opportunity to better understand how repeated efforts, combined with minimal recovery both on the same day and across consecutive days, affect players’ perceived status.

The main findings showed that the scheduled daytime (morning vs. afternoon matches on the same day) did not significantly affect the match load as measured by the s-RPE, nor did it influence perceived enjoyment. Such findings may be attributed to the consistency in players’ playing time across the two sessions. As most players participated for a similar number of minutes in both daytime matches (Table 2), their perceived exertion remained stable throughout the day. Additionally, as suggested by Botonis et al. [37], the frequent players’ rotation implemented during the water polo matches likely contributed to a more even distribution of their perceived loads. With regard to enjoyment, it can be speculated that players maintained high levels due to the competitive nature of the matches and their intrinsic motivation of participating in a highly competitive event. As previously demonstrated in water polo [23], enjoyment may be more influenced by contextual factors, such as the match outcome, than by players’ physical effort or well-being status.

However, the linear mixed model revealed that daytime did impact players’ perceived recovery status (PRS), with values expressed before afternoon matches (PRS_6h_) associated with scores 1.4 points lower than scores indicated before morning matches (PRS_12h_). This difference may be attributed to the structure of the congested schedule, where afternoon matches often followed a morning fixture on the same day, without any specific post-match recovery interventions. As a result, the recovery window between matches was less than 8 h, potentially compounding fatigue and influencing players’ perceptions due to the cumulative load. Our findings confirm the importance of planning recovery strategies between both congested training and matches, as previously demonstrated in the related literature [38]. In fact, as suggested by Barrenetxea-Garcia et al. [17], water polo coaches should modify their habit of underestimating recovery strategies in their periodization. Moreover, as suggested by the literature, these strategies could easily include specific nutritional interventions tailored for water polo players [39] or active recovery protocols as demonstrated in swimming settings [40]. For this reason, water polo coaches should include evidence-based recovery practices in their coaching philosophy. However, current studies suggest that many coaches still miss the habit of monitoring their athletes, highlighting a gap in the management of players’ recovery [13].

Regarding the analysis of friendly and tournament matches, which presented similar s-RPE and HI values, the linear mixed model revealed a main effect for the enjoyment contextual factor, with significantly higher scores in friendly matches compared to official ones. These findings support the notion that, in water polo, the pressure to achieve results and the significance of match outcomes can negatively affect players’ perceived enjoyment [23]. Specifically, the tournament matches were held at the end of a congested schedule (the last five matches within three days), under elevated stress levels and reduced motivation. By that point, players had already completed seven friendly matches, during which they could experiment with technical skills and play more freely without the burden of the competitive pressure of a tournament match.

In our group, the linear mixed model showed that the daily s-RPE was positively associated with the overall Hooper Index, especially with the fatigue sub-scale, and inversely associated with the PRS_12h_. This result aligns with previous research regarding water polo, which found that players’ perceived recovery tended to be lower due to accumulated fatigue and residual stress from consecutive matches and a preceding intensified training period [18].

Moreover, the analysis of the linear mixed model revealed that the match-day contextual factor significantly influenced the sleep sub-scale scores of Hooper Index, the perceived recovery status (PRS), and the daily session rating of perceived exertion (s-RPE). Indeed, as illustrated in Figure 3, the HI showed a gradual increase leading up to the congested match period, peaking around Day 1. This trend suggests a cumulative load effect resulting from the preparatory phase. Notably, the HI remained relatively elevated and stable throughout the central phase of the congested matches period (Day 1 to Day 7), which can be explained by a state of maintained physiological and psychological strain without full recovery between matches. A pronounced decline can be observed immediately after the final matches (from Day 8 to Day +3), indicating that players began their recovery once the intensive match schedule ended. The substantial main effect on sleep across match days aligns with the existing literature in team sport indicating that sleep quantity and quality may significantly fluctuate based on the player’s daily routine [41]. Specifically, for water polo, a previous study by Botonis et al. [19] suggests that the increased workload alongside the inadequate recovery during congested period affects sleep patterns. Indeed, our results speculate that the cumulative fatigue and workload over consecutive days adversely affected players’ sleep quality. Furthermore, being away from home and sleeping in an unfamiliar setting while travelling likely exacerbated their sleep disturbances [20,23]. The decline noted on specific match days in the Perceived Recovery Scale aligns with previous findings indicating that in water polo, recovery perceptions vary according to accumulated training and match load [19]. The significant rise in the daily s-RPE can be attributed to the cumulative internal load from previous days, particularly during congested schedules where players faced repeated high-intensity demands without adequate recovery time or interventions. The progressive increase in load can result in increased perceived exertion on match days, despite the external load remaining constant, as indicated by the average playing time in our study.

### Limitations

This study has several limitations that should be acknowledged. The sample size was relatively small and limited to a single youth water polo team (14 players), which may affect the generalizability of the findings. Although perceived values and measures were monitored using validated questionnaires, these rely on self-reported data, which may be subject to individual bias or misinterpretation. Despite a three-week familiarization period and the athletes’ regular use of these scales throughout the season, some variability in subjective responses cannot be excluded.

Furthermore, the absence of external load measures, technical–tactical analyses [42], or specific swimming tests during the data collection period [43] limits a more comprehensive understanding of the physiological demands placed on the players during the congested match schedule. Moreover, no data were collected regarding the training sessions carried out before the research period, which does not allow any comparison between a regular training period and the congested match schedule in terms of players’ internal load and perceived responses.

Finally, this study focused exclusively on male youth players, and future research should aim to address these gaps and explore whether similar patterns emerge under the new 25 m rules [44], as well as among female and elite senior players.

## 5. Conclusions

Conducting research in real-world scenarios is essential to ensure ecological validity and practical applicability. In this regard, our study offers valuable insights into the internal match load, well-being, perceived recovery, and enjoyment of youth water polo players during a congested competition schedule. Over a seven-day period, players participated in twelve matches (averaging 1.7 matches per day), which resulted in fluctuations in s-RPE, recovery, and general well-being. Notably, perceived enjoyment was higher during official matches compared to friendly ones, suggesting that psychological factors can influence players’ perception independently of physical load.

To better support youth players in such demanding contexts, coaches should consistently apply simple monitoring tools like the session-RPE method, the Hooper Index, and PRS questionnaires. These tools can provide real-time feedback to detect signs of accumulated fatigue or insufficient recovery. Coaches are also encouraged to incorporate brief psychological interventions and tailored recovery strategies, particularly when matches occur less than 6 or 12 h apart for several consecutive days. By following these guidelines, such an approach not only helps optimize short-term performance but may also contribute to safeguarding long-term players’ development in youth water polo.

## Figures and Tables

**Figure 1 sports-13-00286-f001:**
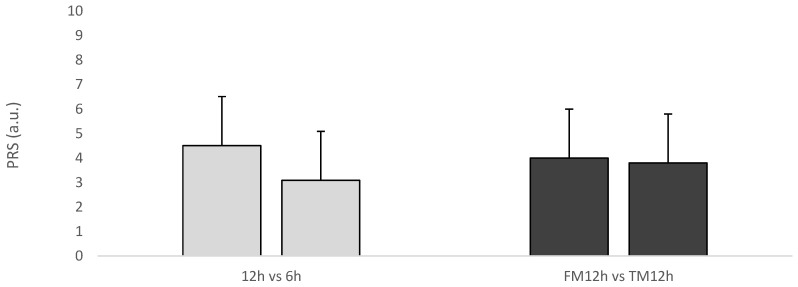
Comparison of PRS values between PRS_12h_ and PRS_6h_ sessions and between PRS_12h_ of friendly matches and tournament matches. Note: PRS = perceived recovery scale; 12h = PRS_12h_; 6h = PRS_6h_; FM12h = PRS_12h_ after friendly match; TM12h = PRS_12h_ after tournament match.

**Figure 2 sports-13-00286-f002:**
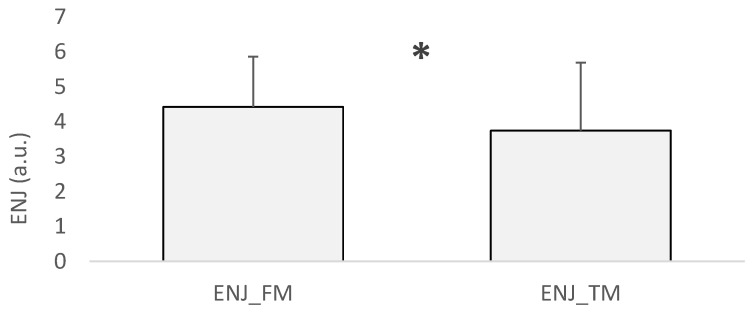
Comparison of ENJ values between friendly matches and tournament matches. Note. FM = friendly match; TM = tournament match; ENJ = Enjoyment Likert Scale questionnaire; * main effect showed by the linear mixed model (*p* = 0.003).

**Figure 3 sports-13-00286-f003:**
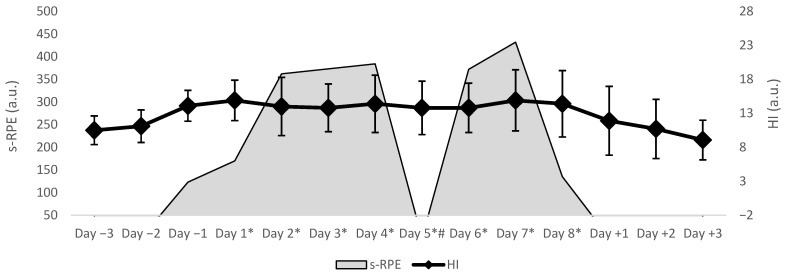
An overlay representation of the s-RPE and Hooper Index scale across the congested period of matches. Note: s-RPE = session-RPE; HI = Hooper Index, * = days of a congested match schedule (data collection period); # = rest day.

**Table 1 sports-13-00286-t001:** Overview of congested match schedule and timeline of monitoring questionnaire administration.

	TUE—day −1	WED—day 1	THU—day 2	FRI—day 3	SAT—day 4	SUN—day 5
	18/07/23	19/07/23	20/07/23	21/07/23	22/07/23	23/07/23
***a.m*.**		HI + PRS_36h_	HI + PRS_12h_	HI + PRS_12h_	HI + PRS_12h_	HI + PRS_12h_
			FM2	FM4	FM6	
			RPE + ENJ	RPE + ENJ	RPE + ENJ	
***p.m*.**			PRS_6h_	PRS_6h_	PRS_6h_	
	R	FM1	FM3	FM5	FM7	R
		RPE + ENJ	RPE + ENJ	RPE + ENJ	RPE + ENJ	
	MON—day 6	TUE—day 7	WED—day 8	THU—day 9	FRI—day 10	SAT—day 11
	24/07/23	25/07/23	26/07/23	27/07/23	28/07/23	29/07/23
***a.m*.**	HI + PRS_36h_	HI + PRS_12h_	HI + PRS_12h_	HI	HI	HI
	TM1	TM3	TM5			
	RPE + ENJ	RPE + ENJ	RPE + ENJ			
***p.m*.**	PRS_6h_	PRS_6h_				
	TM2	TM4		R	R	R
	RPE + ENJ	RPE + ENJ				

Note. HI = Hooper Index questionnaire; PRS = Perceived Recovery Scale questionnaire; FM = friendly match; R = Rest Day; RPE = Rated Perceived Exertion; ENJ = Enjoyment Likert Scale questionnaire; TM = tournament match.

**Table 2 sports-13-00286-t002:** Descriptive statistics of dependent variables (mean ± SD).

	Mean	SD
RPE_FM (a.u.)	6.72	1.70
RPE_TM (a.u.)	6.69	2.46
ENJ_FM (a.u.)	4.42	1.40
ENJ_TM (a.u.)	3.74	1.95
s-RPE_FM (a.u.)	177	80.8
s-RPE_TM (a.u.)	183	97.5
Players’ playing time_a.m. (min)	25.3	8.89
Players’ playing time_p.m. (min)	25.5	8.29
Players’ playing time_FM (min)	25.6	8.23
Players’ playing time_TM (min)	25.1	9.08

Note. RPE = Rated Perceived Exertion; FM = friendly match; TM = tournament match; ENJ = Enjoyment Likert Scale questionnaire; s-RPE = session-RPE; a.m. = morning match; p.m. = afternoon match.

## Data Availability

Data are contained within the article.

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
