# Peer review of "Impact of a Congested Match Schedule on Internal Load, Recovery, Well-Being, and Enjoyment in U16 Youth Water Polo Players"

_sports, 2025, doi:10.3390/sports13090286_

Round 1

Reviewer 1 Report

Comments and Suggestions for Authors

The article discusses fatigue among young water polo players during the competitive cycle.

As the authors rightly point out, there is a lack of articles on the perception of fatigue among youth players, both in water polo and other sports. It is worthwhile to focus on analyzing exertion and fatigue in youth players to better support senior national teams. This topic is crucial for coaches of youth teams and organizers of official, classified competitions.

This article relies on subjective assessment sheets such as RPE, s-RPE, ENJ, PRS, and HI. In this type of research, it is important for players to be familiar with these sheets, so they can progress from level 0 to level 10. Many youth players do not distinguish between levels on the Borg scale, often underestimating or overestimating their scores depending on the importance of the match (official vs. unofficial) and whether the match was won or lost. Fatigue levels are often dictated by off-field situations, so it is worth adding whether players were at a training camp or returning home during the training cycle. According to the authors, the athletes were familiar with the RPE sheets, but it is unclear whether they perform this assessment daily or if it was a one-time occurrence.
In the introduction, the authors focus heavily on reports from other authors. They state that fatigue is diagnosed, but in my opinion, the authors should focus on explaining the sheets and indicators used (ENJ, PRS, HI), their reliability and repeatability, their purpose, and the role of fatigue based on available scientific articles. The introduction, especially paragraphs 2 and 3, should be redrafted. There is a wealth of information on s-RPE in available publications – the authors should add several citations in this area confirming the importance of this indicator for research on young athletes.
The authors should avoid using the "-" symbols – lines 101, 117, 150, and 151.
In chapter 2.2 Subjects – please remove the club name (S.S. Lazio Nuoto) and, where possible, include the athletes' experience. Please complete section 2.3 Procedures with the following information: 1) whether fatigue reports were sent individually or in groups. Was the 30-minute time limit mentioned for completing the forms, or were the forms sent after 30 minutes? This is quite important for the reliability of the collected data.

Section 2.4. Poorly described. Please add the assumptions (whether the assumptions were met or not met) for applying the given statistical test. Is it appropriate to rely on means in the given group, or is it appropriate to use nonparametric tests based on the median? What was the effect size? In the results section, it's worth presenting daily data (as in the case of the (HI - index). If no recovery programs were used, but only fatigue assessment, it's worth adding information such as the increasing fatigue that developed after friendly and league matches from day to day.
Please also describe in detail the "mental shortcut" <X day, -X day (lines 56, 58, 63, 91, 94, 95).
The conclusion contains answers to the research questions. While reading the article, I was hoping to indicate fatigue levels in friendly and league matches, as well as during the training cycle. This chapter is a culmination of the player observations, but also provides tips and recommendations for further observations and coaching of players in such match cycles.
The authors mentioned limitations. These should be described in a separate chapter.
I hope my tips will help you better describe your observations.
Best regards

Author Response

Dear Reviewer 1,

We would like to sincerely thank you for your thoughtful and constructive feedback. Your comments have been extremely valuable in improving the clarity, rigor, and overall quality of our manuscript. We carefully addressed each of your points and revised the manuscript accordingly. All changes have been marked using track changes in the revised version. Below, we provide detailed responses to each of your comments, following the same order in which they were presented.

Reviewer 2 Report

Comments and Suggestions for Authors

General Comment

The authors provide a study monitoring internal load, well-being and recovery status in elite U16 16 male water polo players during a congested competitive period. Although there are good aspects here, I think that some limitations arise and may impair the results interpretation and scientific soundness. A thorough revision of the manuscript and a new evaluation will be essential before further proceedings.

Specific Comments

Lines 52-70. The way you expose the literature by spending almost three sentences on each study is quite strange and denotes a resemblance on how AI makes study interpretation. In one paragraph, comprising two sentences, you can consolidate the information from the various studies comparing regular versus congested training weeks. This is my humble opinion.

Line 72-73. What do you mean by “tactical-periodization approach”? Periodization and planning are not previously defined? I think you should rephrase this.

Line 100-103. Although I understand that this can happen, this is not the common trend in some countries. So, some reservations should be made about the water polo teams having the same amount of congested weeks than remaining team sports mostly at younger ages like U-16.

A hypothesis is missing at the end of your study.

Probably you should provide first the “subjects” and then the “experimental design”.

Line 162-163. Why measure RPE just after the match? Don’t you think that players who ended the match in the bench would respond differently from the others? With only 14 players, this probably may have affected your results.

Line 168. What PRS means?

How can you ensure that you recruited “elite” players? Those are U16 players, and nothing is clear that they are or could reach an elite level. Nothing is stated about any kind of classification to characterize those as elite.

Line 180. “Weight” should be “mass”.

Line 197. An “l” is missing in “players”.

Your are measuring perceived recovery and enjoyment. Don0t you think that those are more psychological issues than internal load measurements? I think that this could lead to revise your title and related issues.

Your methods include delivering to the players respond to questionnaires in a non-controlled environment. I think that there is a great limitation here. Moreover, sleep monitoring could be even more accurate with other kinds of devices which leaves me the doubt is recovery issues and sleep quality where really well assessed.

At the end, you just analysed the congested training period. Your discussion is well written and covers the topic in articulation with literature. However, how can you ensure that this could be something different from non-congested training periods? How can you conclude that coaches should use recovery strategies since you have not compared with other kinds of training types to see if they are better or worse at recovering? Indeed, you sometimes spend speculating beyond your findings, which is not good when doing science. So, you should rewrite some of those sections.

For me, a section of “limitations” is missing. Probably, lines 386-393 could be transferred and pasted at the end of your discussion.

I think that you should be more precise in your conclusion. From the way it is written, it is not understandable what you conclude from your work.

Author Response

Dear Reviewer 2,

We appreciated your time and effort dedicated to providing feedback to our manuscript and we are grateful for the insightful comments on and valuable improvements to our paper. We have followed all your suggestions and changes are highlighted in track changes in the re-submitted files. In this document, we are going to answer your questions in the same order you send them to us.

Reviewer 3 Report

Comments and Suggestions for Authors

Dear Sports editorial team, thank you for inviting me to review the article entitled “The impact of a congested match schedule on the internal load response of elite U16 water polo players”. The aims of the article were the monitor the internal load, well-being and recovery status of elite U16 male water polo players during a congested competitive period.

Overall, the article is insightful and proposes a meaningful question. Certainly, understanding the demands of congested schedules will be of benefit when planning performance strategies. The paper offers insight into the concerns of a congested schedule and potential risks. However, there is a lack of use of the data collected and practical implications of the study. The authors claim such scales and tools are beneficial but do not demonstrate the utility of doing so in the article. There are a number of questions and concerns I must raise that I have articulated in the following comments.

General feedback

The article is written well, however there are a number of errors throughout that must be addressed before publication. The sample size is small and potentially underpowered. That being said, the authors note that one team was included and this limits sample size. This must be recognised in limitation as do a number of other limitations that this study has. There is no formal recognition of limitations in the article which I recommend be included. The major concern is with the details concerning statistical analysis and lack of formal analysis. The authors claim that it is important to manage and monitor these variables, however, the study show no real world practical utility of the variables, such as showing how RPE and HI can be used, is there a relationship between increased perceived load and HI? This would certainly be valuable to practitioners looking to monitor player load. This is a trend across most of the variables included in this study.

Abstract

  1. Use the language of your surveys, you use RPE, Enjoyment, and Perceived Recovery, and Hooper index. Yet you refer to a fatigue item in your findings. This must be updated, please report data and descriptors for tools used accurately. Or explain how fatigue was measured accurately.

Introduction

  1. Line 49-52 needs to be reference
  2. Lines 55-72 Understandably explores the current body of literature but forms a discussion that is best reserved for the discussion section. Certainly, it is good to introduce findings to build an introduction using previous research, however, this makes the introduction overly long. This is the same for lines 74-96, I appreciate the authors wish to provide some background, however, the writing style here is not concise.
  3. Please revise the introduction to focus on the question and sporting code at hand, the introduction provides much that should be present in the discussion .
  4. Line 152 please change main purpose to aim of the study

Methods

  1. Please be specific, are the participants youth athletes? Or adolescents? Young is ambiguous.
  2. Line 163 the question is incorrectly worded, why was this changed? The question was validated using a specific phrase, please justify why the question was modified? Or provide a citation to the question.
  3. Did the research team obtain assent from the participants?
  4. Line 197 how long was the familiarisation process?
  5. Line 214 referencing style is different here.
  6. Table 1: Please organise abbreviations in notes as the first appear in the table top left to right.
  7. Please provide evidence that each of the surveys implemented are reliable and valid in the cohort examined.
  8. Much greater detail is required in the statistical analysis section to better express what is being analysed. Daily changes, pre- or post-game, 6hr vs. 12hr, FM vs. TM. Factors derived from the scales used. Did those athletes who experienced great RPE recovery slower? Sleep less? Perceived enjoyment differently? There are analysis in the results that are not explained in the statistical analysis section. Why were 95% CI not included in the reporting? Why was there not consideration of effect sizes? Was there a relationship between RPE and DOMS?

Results

  1. Table 2: Please organise note abbreviations as above. This table would also benefit from player descriptive data.
  2. Figure 1: The standard deviation is presented incorrectly.
  3. Figure 1: If significant, why was the model not explored? Why was PRS6hr not reported between FM and TM?
  4. Figure 2: As above, SD is incorrect.
  5. Figures would benefit from presenting other variables, such as an overlay of RPE and HI.

Discussion

  1. Lines 282 why is sleep now a variable? Throughout the manuscript, various measures from the Hooper Index, this is misleading.
  2. In the discussion the authors discuss an analysis of morning vs. afternoon scheduling, this was not an aim or described in the methods section (please forgive me if I missed anything). This is outside the scope of the article at present and should be included formally, or not.
  3. Lines 318 is not written well. The authors make a recommendation, that is supported by a reference. This is not articulated well and is a trend in the discussion.
  4. Lines 327 onwards, again, this analysis is not included or described in the methods and must be formally provided in the methods section and statistical analysis.
  5. There is a lack of recognition of limitations that is required.

Author Response

Dear Reviewer 3,

We deeply thank you for your valuable suggestions and comments on our manuscript. Those comments are of enormous assistance to us for improving and revising our manuscript. We have studied the comments carefully and made corrections in line with the suggestions you made. We have marked the revision in red (and highlighted in track changes) in the re-submitted files and included our responses to your comments below.

Round 2

Reviewer 2 Report

Comments and Suggestions for Authors

General Comment

The authors provided reasonable changes according to my previous opinion. I just have few comments.

Specific Comments

Thanks for your effort in writing a hypothesis. However, I will suggest matching the hypothesis with the aim of your study. If you have just one sentence for the aim you should have one sentence for the hypothesis as well.

Once again, I will ask you to first provide the subject's description and then the experimental design. This is not my way of thinking, but, instead, is the way that better provides the rationale without duplicating information. As an example, in the “experimental design” you are referring to some procedures used in data collection (like RPE and questionnaires) which are again mentioned in the “procedures” subsection. So, please revise.

I still think that no U16 athletes, despite of the sport, should be characterized as “elite”. Since you do not have any kind of classification or reference that helps you to support that characterization, I suggest removing the word “elite” throughout the text, as it is not presented in the title.

Author Response

Dear Reviewer 2,

We would like to sincerely thank you for your feedbacks. We carefully addressed each of your points and revised the manuscript accordingly. All changes have been marked using track changes in the revised version. Below, we provide detailed responses to each of your comments, following the same order in which they were presented. Thank you again

Reviewer 3 Report

Comments and Suggestions for Authors

Dear Sports Editorial Team, 

Thank you for inviting me to re-review the article entitled “Impact of a Congested Match Schedule on Internal Load, Recovery, Well-Being and Enjoyment in U16 Youth Water Polo Players”. As previously stated, I believe the article is warranted. I have reviewed the authors edits and responses to my previous comments and recognise the efforts made to address all comments made.

Please provide a reference for Cohen’s d interpretations.

The article is as the authors suggest, improved. I have very few suggestions before I would suggest acceptance. Figure 1 and figure 2 appears to only show one direction of error (error bars only show positive direction). This should be revised if necessary to reflect the full range of error.

Within your limitations, the paragraph layout differs from the approach you have previously adopted.

Lastly, there remains several errors in the consistency of your reference list, for instance, bolding the date is incorrect in reference 5. Please ensure all references are correctly presented before publication.

Author Response

Dear reviewer 3,
Thank you for this second revision, which will improve the readability of the paper.
We will respond point by point in this file, and all responses have been edited into the manuscript.
Thank you!
